# Self-Administration of a Boar Priming Pheromone Stimulates Puberty in Gilts without Boar Exposure

**DOI:** 10.3390/ani14010091

**Published:** 2023-12-27

**Authors:** John J. McGlone, Lauren Duke, Maya Sanchez, Arlene Garcia

**Affiliations:** 1Laboratory of Animal Behavior, Physiology and Welfare, Texas Tech University, Lubbock, TX 79409, USA; 2School of Veterinary Medicine, Texas Tech University, Amarillo, TX 79106, USA

**Keywords:** pigs, gilts, estrus, behavior, pheromone

## Abstract

**Simple Summary:**

All commercial swine farms require replacement gilts. Gilts are grown in pens and are commonly exposed to boars to induce estrus. Boar exposure to peri-puberal gilts is time-consuming, with variable results on commercial farms. Our goal was to determine whether a novel boar pheromone (Boar Better^®^; BB) could replace a live boar, accelerate the onset of puberty, and obtain equal or better percentages of gilts bred. A novel sprayer that allowed gilts to self-administer BB (a putative boar priming pheromone for gilts) at will. The sprayer is an environmental enrichment (EE) device that pigs root, push, and chew. The EE device can all gilts to self-deliver BB during the late pre-puberal period, which may save labor and improve worker and gilt safety by reducing the use of a live boar. In this work, gilts were exposed to either boars on a daily basis or BB self-administered in the EE sprayer. In conclusion, BB induced the same or more gilts to be identified in estrus compared to live boar exposure, and injured fewer gilts while reducing the labor and boar maintenance costs. The data support the idea that BB acted as a priming pheromone to stimulate the onset of estrus in gilts without a boar.

**Abstract:**

Labor is in short supply in animal agriculture. One time-consuming task is estrus detection in gilts. Stimulation with a live boar causes the onset of puberty in young gilts. Typically, a live boar is used to stimulate and identify estrus in the gilts by exposing the gilts to him. Recently, a boar pheromone (BB) was developed to replace the use of a live boar for sows. Additionally, a novel automatic sprayer used as environmental enrichment (EE) by gilts for the self-administration of BB has been developed by this laboratory. A commercial study was conducted to determine whether the use of a live boar could be replaced with a simple EE sprayer, allowing gilts to self-administer BB. Our objective was to determine whether the number and percentage of gilts in estrus obtained using live boars was comparable to self-administration using an EE sprayer containing BB. A total of 242 gilts were randomly assigned to either a live boar (BOAR) or BB self-administration using the environmental enrichment (EE) sprayer. Gilts began simultaneous exposure to either the BOAR or the BB when they were about 4–5 months of age and this continued until they were found in estrus or were injured, died, or never cycled about 2 months later. A total of 83.3% of gilts with exposure to BOAR were identified in estrus and bred, while exposure to BB resulted in 92.9% of gilts reaching puberty and being bred (*p* < 0.05). The days to reach estrus were 11 days longer for gilts exposed to BB than BOAR. Eight percent more gilts were injured by the BOAR than by using BB (and no boar). The use of BB as a priming pheromone could prevent gilt injuries, save labor, and reduce costs for pig farmers while not inhibiting reproductive output.

## 1. Introduction

Commercial sow farms require the introduction of new genetics through replacement gilts. Each year, 40–60% of sows are replaced with gilts on modern commercial farms [1]. Gilts often develop in gilt development unit barns like finishing barns. When gilts reach a target weight (over about 110 kg) and are about 5–6 months of age, a common practice is to expose them to a live boar to stimulate the onset of estrus [1]. The goal is to use the boar’s sight, sounds, touch, behavior, and olfactory signals to trigger the onset of estrus. This procedure is called boar exposure. When boar exposure is performed correctly, a high percentage of gilts will reach puberty and be bred on their second or third estrus. However, boar exposure is variable depending on how long the person handling the boar exposes him to the gilts, as limited exposure can prevent the stimulating effects of the boar. Boars are usually exposed to the gilts once per day. Handling boars is a safety concern, as they can injure people or animals. The use of a boar requires significant human labor, since boars need to be walked in and out of gilt development units.

Boar exposure has been repeatedly shown to bring on gilt estrus [2]. In one report, daily exposure to a boar induced 88% of gilts to express estrus, while only 46% of gilts without boar exposure expressed estrus [2]; this phenomenon is well known among pig breeders.

Boar Better^®^ (BB) is a novel boar pheromone that contains three active molecules (androstenone, androstenol, and quinoline) and is a commercial product that has been used to stimulate reproductive behaviors in recently weaned sows [2,3]. In this sense, BB acts as a releasing pheromone to stimulate sexual behavior after the sow smells the pheromone [4]. Gilts require a priming pheromone to prime their hormonal and reproductive systems for the onset of puberty. Rather than a single spray, priming pheromones often require frequent exposure of the olfactory signals to accommodate the priming effect, which could take days or weeks. Until now, BB has not been shown to be a priming pheromone to stimulate the onset of puberty in gilts.

Prior to this field study, we developed an environmental enrichment (EE) sprayer with which gilts could self-administer any liquid at will [5], such as vaccines and pheromones. In earlier work with sows, we found that pregnant sows would use the sprayer one to two times per day on average, whereas sows in estrus averaged 17 sprays per day (unpublished work in preparation). Thus, it is likely that gilts would utilize the sprayer when in estrus similar to sows. The BB pheromone was combined with a green dye, and therefore any animal that used the EE device and self-administered the BB frequently would have green dye on their face. Based on the sow behavior, we hypothesized that gilts in estrus would be easily identified as in estrus due to a large amount of green dye being on their face even without boar exposure. Our objective was to compare the rates of estrus detection and breeding on a commercial farm when gilts had either live boar exposure once per day (the farm’s standard practice) or an EE self-administering sprayer with BB. If successful, this tool could be a labor-savings methodology in the identification of a gilts’ first or subsequent standing estrus behaviors. From a biological perspective, BB or other pig pheromones have not been shown to be priming pheromones in domestic pigs. Our broad goal was to determine whether BB can act as a priming pheromone, and more specifically whether we could achieve similar reproductive output with less labor when BB was self-administered instead of administered via exposure to a live boar.

## 2. Materials and Methods

The Texas Tech University (TTU) Institutional Animal Care and Use Committee (IACUC) approved this work prior to the start of the work (IACUC approval number 21044-04). The work was performed on a commercial farm. It was a 5000-sow breeding, farrowing, and nursery farm that sells pigs for breeding. The farm used AcuFast Genetics to produce Vantage gilts (https://www.acufastswine.com/; accessed on 25 December 2023). The farm was a genetic multiplication farm producing commercial gilts for other farms. The genetics on this farm are the parents of the commercial gilt. Gilts are selected for heat check and breeding at about 5.5 months of age, when boar exposure begins. Young (8–24 months of age), vasectomized boars were used as the heat check boars. The boar exposure involved walking a live boar into each pen of peri-puberal gilts with the goal of having the boar interact with each gilt to see whether estrus behaviors (standing still with back pressure and interest in the boar) were present. The once-per-day boar exposure happened in the morning and afternoons of each day.

The farm’s goal was to produce 200 gilts per month as replacement females. The farm used electronic sow feeders (ESF) for bred females. The peri-puberal gilts were housed in pens of about 50 gilts each and fed a fortified commercial diet and water ad libitum. Daily boar exposure and mating was during a 60-day period. The standard protocol (the control group for this study) used boar exposure the first 30 days and their first estrus was intended to be recorded. Then, they were moved into ESF training group pens, where gilts had their second estrus and boar exposure was continued. The gilts were then bred on their second or third estrus and moved to the sow herd.

In the control treatment, the gilts were exposed to boars (BOAR) once per day, at variable times, but most often in the morning, in their home pens. The boar was walked around the pen in an attempt for him to interact with each gilt. If a gilt showed signs of estrus, she was tagged and the estrus date noted (she was identified as being scheduled to breed on her 2nd or 3rd estrus). For the treatment group, the gilts had two EE sprayers per about 50 gilts per pen. They were not exposed to boars in their pen, but boars were present in other pens and in the barn. Workers checked and added BB to the pens each day if needed. A 2 L bottle used in the EE sprayer lasted about 7 days for a pen per 25 gilts (we used 2 sprayers per pen of about 50 gilts). The average BB use was about 11–12 sprays (40–50 mL) per day; however, this is deceiving because some gilts were in estrus and used it 20 or more times per day, while the gilts not in estrus used it only once or twice per day.

Earlier work has shown that each molecule in the BB pheromone (androstenone, androstenol, and quinoline) stimulates different parts of sow sexual behavior, while when combined, a full behavioral repertoire of standing estrus behavior is expressed by sows [2,3,4].

The EE sprayer contained BB combined with a green dye and was hung on the fence of each gilt pen (Figure 1). The EE sprayer had no external plumbing or electricity. It operated mechanically by the gilt pressing on the front panel. If the push was strong enough, a 4 mL spray was delivered. The bottle containing the BB pheromone was attached to a nozzle with a spring and a trigger handle. When the trigger sprayer was pressed, the 4 mL liquid was delivered. The functionality of the EE sprayer has previously been used to deliver an oral vaccine [5]. BB sprayers were also available in the training pens of the BB treatment group. The training pens also included boar exposure for the BOAR treatment group.

This simple study had two treatments and evaluated 242 gilts. The gilts were the experimental units. Most data were reported as percentages and thus non-parametric Chi-square analyses were performed to examine the treatment differences. Measures included the % of gilts that expressed estrus behaviors sufficient to be bred. Some gilts were culled for reasons given in the results. The raw count data are reported so that the reader can make additional calculations that might be of interest.

## 3. Results

The measures of gilt reproduction are given in Table 1. Gilts were culled for several reasons including being unsound or injured, or having died (the cause of death or injury was not reported, but the farm said the rates of injury and death were typical of this system). Data are available for 91 gilts exposed daily to a boar and 151 gilts in pens with two EE sprayers containing the BB pheromone. The gilts used about 8 L per 50 gilts over a 30-day period. This averages 160 mL/gilt or about 40 sprays over the 30-day period. This observation is a rough estimate because some bottles leaked and we did not have accurate use data. However, all gilts had green dye on their face, indicating that they used the sprayer especially when they were in standing estrus.

A total of 20.9% of the BOAR-exposed gilts were culled, while 16.6% of the BB-exposed gilts were culled. When we consider only those gilts that were unsound, injured, or dead, the BOAR-exposed gilts had a loss of 12.1% of gilts, while the BB gilts lost only 4.0% of gilts due to these causes (*p* < 0.05). Gilts that were not culled were considered eligible to be bred. Of the gilts eligible to be bred, 83.3% of the BOAR-exposed gilts were identified in estrus, while 92.9% of BB-exposed gilts were in estrus (*p* < 0.05). Both the control (BOAR) and BB-exposed gilts had reasonable rates of estrus onset. Both the live boar and the BB induced similar rates of estrus in the peri-puberal gilts.

The gilts in the BB-exposed group passed their first estrus 11 days later than BOAR-exposed gilts, yet this time difference was not significantly different. If BB exposure resulted in a later onset of puberty, a large sample size would be needed to confirm this idea (or not).

## 4. Discussion

This is the first demonstration that anything other than a live boar can stimulate the onset of puberty in gilts. However, several key findings from this work are worthy of further study. The fact that over 90% of gilts cycled with BB alone with no boar exposure is a meaningful finding. The percentage of gilts reaching estrus of those eligible was higher for the BB than BOAR-exposed gilts. One possible explanation for this is that the gilts self-exposed themselves to BB many times per day. It could be that multiple boar pheromone exposures are needed to bring on higher rates of estrus in gilts [2]. In any case, the data support the idea that BB is not just a releasing pheromone (the male odor releases sexual behavior in sows) but BB can also act as a priming pheromone (stimulates the onset of estrus without a boar).

This work was a controlled field study in a commercial setting. It would be better to have the two treatments in different air spaces, but this was not possible. Therefore, some BB gilts may have experienced some boar odor (for example, when they walked by in the aisle). Likewise, the boar-exposed gilts would have been able to smell (at low concentrations) the BB from neighboring pens. In spite of these limitations, treatment differences were observed. We have measured androstenone away from point sources and, although it is volatile, its concentration drops further from a point source (like a boar). The concentration a few meters away from a boar may not be high enough to elicit estrus stimulation. Indeed, boar exposure (in-pen) stimulates more gilts to reach estrus than fence-line contact alone [6,7,8,9]. Thus, having live boars in the same room may not have impacted BB-treated gilts because these gilts did not have direct boar exposure.

The relatively high rate of gilts being removed due to injury, death, or being unsound (e.g., problems with their feet structure) was surprising. A total of 12% of BOAR-exposed gilts were removed due to injuries or being unsound, so one must ask whether the boar exposure is causing injury to gilts. The rate of injured, unsound, and dead (IUD) gilts was three times higher when live boars were used than when BB was used. Humans are also injured by boars at times [7], but especially during boar movements, as during heat check. If an alternative to live boar exposure were identified, gilt welfare and human safety may be improved.

Our results may encourage people to attempt to stimulate estrus with pheromone therapy rather than with a live boar. The EE sprayer becomes critical in this approach because if one had to spray gilts by hand, the use of BB would be very time-consuming. When workers arrive to gilt barns with EE sprayers with BB, they immediately see some gilts with green faces (the dye in BB is green), and they can confirm they are in heat by either pressing on their back to illicit the standing reflex or they can bring in a boar to initiate lordosis in the gilt. By not having to run a boar past each animal, time is saved.

A critical feature of the EE self-administration sprayer is that gilts have an opportunity to apply the BB pheromone at will. The pens used an average of 2 L of BB per week per 25 gilts. This averages about 11–12 sprays per gilt per day (each spray is 4 mL). When a gilt was not in heat, they used the sprayer 1 or 2 times per day. When they were in estrus, they used the sprayer 12–20 times per day.

Two possible explanations are hypothesized to explain BB-treated gilts having a higher rate of estrus detection than BOAR-exposed gilts. First, the estrus may have been delayed among BB-exposed gilts, which may have allowed for more reproductive tract maturity. Second, gilts are more likely to be stimulated with multiple pulses of pheromone than a single exposure to a boar once per day. Because they are self-applying the pheromone, they can have as much or as little as they want. This brings up the need to find the right dose and the number of sprays over time of the BB pheromone to best prime gilts for their first estrus.

## 5. Conclusions

The use of EE with the BB pheromone stimulated estrus in gilts at the same rate as or a higher rate than a live boar. The use of the BB pheromone may take longer to trigger the onset of puberty (by 11 days) than the use of a live boar for gilts to come into heat. However, running a live boar among the gilts seemed to cause a higher rate of injured, unsound, and dead gilts. The use of the BB pheromone may be useful in stimulating the onset of estrus in gilts with or without the use of a live boar. Labor can be saved when gilts use EE to self-administer the BB pheromone rather than a person walking a boar into each pen. The rate of gilts identified as having a first estrus was equal or better with BB than when a live boar was used.

## 6. Patents

All patents related to this project are owned by Texas Tech University, with J.J.M. as the inventor and assigned to Animal Biotech (Dallas, TX, USA). BB has an issued patent. The sprayer is currently patent-pending. The Boar Better® USA patent is 9,480,689, (1 November 2016) and its EU patent is EP3724351B1.

## Figures and Tables

**Figure 1 animals-14-00091-f001:**
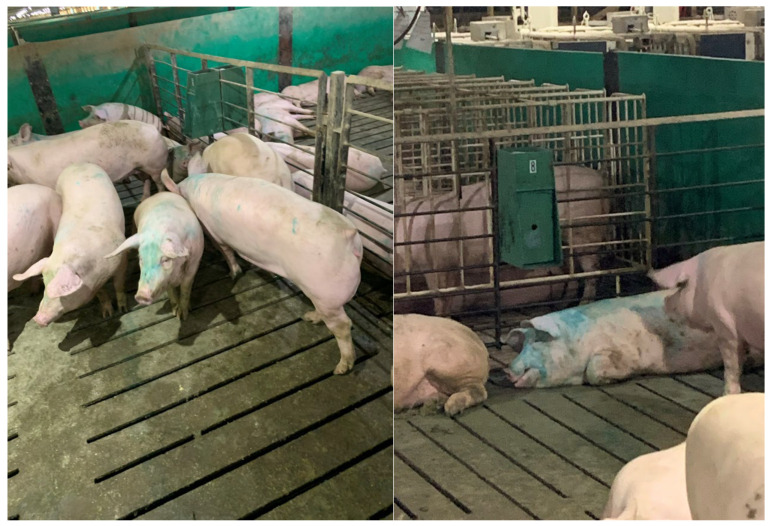
Photo of gilts in an experimental pen. (**Left**) Note the one gilt facing the camera that is in estrus and has green dye on her forehead. (**Right**) Note the gilt lying under the sprayer with green dye indicating she is in estrus (she has self-applied BB from the green sprayer). This gilt also stayed close to the sprayer when she was in full standing estrus. The sprayer works via the pig rooting and pushing the device, which delivers a 4 mL spray of BB. These pens of gilts did not have boar exposure as the control pens did.

**Table 1 animals-14-00091-t001:** Measures of gilt reproduction. N is the number of observations. Eligible to be bred gilts are those that are structurally sound and healthy. IUD is the number or percentage of gilts that were not eligible to be bred because they were either injured or unsound or they died. *p*-values are only presented when significant differences were found. Percentage data were analyzed using Chi-square analyses. In the *p*-value column, where no *p*-value is shown, this indicates either that it is not applicable or the effect was not significantly different (*p* > 0.10).

Measure	BOAR Control	BB	*p*-Value
N gilts	91	151	
N culled	19	25	
% culled	20.9	16.6	
N eligible to be bred	72	127	
N bred	60	118	
% bred of eligible	83.3	92.9	*p* < 0.05
% bred of those started	65.9	78.1	*p* < 0.10
% IUD (injured, unsound, died)	12.1	4.0	*p* < 0.05
Average days to estrus after boar or BB exposure	23.4	32.6	

## Data Availability

The primary data are in the text of the paper.

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
