# Peer review of "Self-Administration of a Boar Priming Pheromone Stimulates Puberty in Gilts without Boar Exposure"

_animals, 2023, doi:10.3390/ani14010091_

Round 1

Reviewer 1 Report

Comments and Suggestions for Authors

General

Paper of McGlone at all.  Sent to Animals as “Communication” concerns an application of novel boar pheromone preparation Boar Better (BB) to replace the use of live boar to induce puberty in gilts. Additionally, a novel automatic sprayer used as environmental enrichment by gilts for self-administration of BB was  applied.

The manuscript is quite interesting but my doubt is whether it can be published as scientific paper or rather as “case report” since given methodological conditions of experiment can not be repeated (reproduced) in other porcine farms.

The treatment group gilts treated with BB plus green dye were not exposed to boars in their pen directly but boars were present in other pens in the barn and gilts could have visual, sounds and smelling (odor ) contact. This specific farm effect could be decisive for results . To elucidate the potential boar presence effect and to separate it from the  pure BB pheromone effects on the onset of puberty this research project  should be extended for two additional “control” groups : one including prepubertal gilts alone and the second consisted of the  prepubertal gilts challenged with BB. Both groups should be however close separated from any potential live boar influence .

Specific

A procedure of gilts exposure to boars is not adequately described. How long the boar was exposed to gilts and in what time of the day?

According to the literature androstenedione and androstanol are releasing pheromones. The authors claim that they  can also act as priming pheromones (l. 173 -175) .Is it proved ?

l. 18  What means “  Daily sue “

l. 19 should be.. ( and no boar)

Comments on the Quality of English Language

Author Response

Thank you for the review.  The study is what we call a field study; it is more than a case study.  However, if this journal has a designation of "Case Study" then calling it that would be fine with us, although not preferred.  To us, a case study typically involves a single animal or a single case.  This study has sufficient replication to allow analyses.  And it examines 2 treatment groups.

As a practical matter, it is difficult for a university farm to do studies with hundreds of gilts.  And if we did a university study, one would still wonder how it might work in the field.   

We also agree that gilts in the BB treatment group would have fence-line exposure to boars as they walk by or as they are briefly in neighboring pens.  We added text about this to the discussion.

All of the other changes were made; typos were fixed; more was added about boar exposure, and we point out in the discussion what the reviewer suggested (that BB and BOAR treatments would cross-contaminate each other at very low concentrations).  

The reviewer asks if these data "prove" that BB is a priming pheromone.  First, it is rare that any single study can prove anything.  Second, we do not state that we have proven that BB is a priming pheromone, rather we state that the data support the hypothesis that BB is a priming pheromone.  We agree more data from different studies would add confidence to eventually prove that BB is a priming pheromone. 

Reviewer 2 Report

Comments and Suggestions for Authors

The manuscript meets the scope of the journal and requirements for Communication. The topic is very interesting and expands the knowledge about the use of oestrus detection methods in pigs and it suitably complements the results of these authors published earlier in 2019.

Title page: The title of the article is clear and informative.

Simple Summary: I see it too long. I suggest shortening it by deleting the part with concrete results – lines 17 to 22, starting with Gilts with boar exposure…. ending with In conclusion,. L17 – self-administered.

Abstract: The abstract is clear including all parts of traditionally accepted article abstracts.

Introduction: The introduction widely discusses the problem of the use of live boars and their pheromones in estrus detection in female pigs. Since Boar Better is a patented commercial product, I suggest using the mark of this product as BOARBETTER® as you published previously in Animals 2019.; L63 - …pheromone product..

Material and methods: The methods used are appropriate for obtaining significant data. However, I have a few questions and suggestions for improvement. L92 – …(IACUC approval number; 21044-04); L97 – State the breed, live weight, and mean inter-estrus interval in gilts. State the breed and live weight of the boars used.; L101 – How long did the boar exposure continue? What does it mean the limited boar exposure? What was the frequency of exposure?; L103 – Were the gilts exposed to boars regularly at the same of the day?; L106 – explain the abbreviation when mentioned for the first time in each main section of the article; L107 – Since the boars were present in the same barn, although not in the same pen as gilts, their pheromones still stimulated the gilts, in addition to the sprayers and this fact should be included in the explanation of the study results.    

Results: Results are clear concerning their tabular and graphical representation. L144 – delete a comma after including; L145 – delete a space after of.

Figures: I suggest the images be shown closer to each other – put smaller space between them; L127 - Note

Tables: L165 – Why the not significant difference is stated at P˃0.10 when the least not significance is supposed to be at P˃0.05?; in Table 1 – Should not a P-value of PË‚0.10 be PË‚0.01? Even though it looks insignificant, in my opinion.

Discussion: This part is deeply discussed with the relevant available literature. L180 – explain IUD; L182 – identified.

Patents: Please, state the number and title of the BB and pending EE product patents.

References: References used are formally appropriate. Just put the name of the journal and volume in italics and number the order of the references correctly.

I hope my suggestions will aid in improving this interesting manuscript. I recommend the manuscript for publication after a minor revision.

Author Response

The simple summary was reduced in length as suggested.

The ® was added when the product name was mentioned.

Details on the breeds, weights and inter-estrus interval were added to the methods.  A reference to the genetic company that sells this genetic line of pig is included.  L 86-93.

More details were added in the methods about boar exposure as requested by the reviewer.  Issues about BB-treated gilts having some fence-line boar exposure were discussed in the revised discussion.

Results: Typos fixed.  Images were placed closer, however, final galley proofs should straighten this issue. 

We call this a trend; when P < 0.10 but not P < 0.05.  The % eligible to be bred and % injured, unsound or died was greater for Boar than BB treatment groups – these 2 are statistically significant. 

All other editorial changes were made as the reviewer suggested.

Reviewer 3 Report

Comments and Suggestions for Authors

1. The green dye can be seen if the gilts use the EE sprayer. How to count the times of a gilt using the sprayer per day?

2. Line 16, “BB self0administered”?

3. Line 127, not or note?

4. The format of references should be modified according to the journal guidelines.

Author Response

Video cameras recorded gilt/sow behavior in other studies of ours.  We did not have video cameras here.  We estimated the number of sprays based on the usage or disappearance of BB from the bottle.  This is only a rough estimate.  We added some text in the results at the end of the first paragraph indicating use of BB in the sprayers.

All other editorial suggestions were made as suggested by this reviewer.

Round 2

Reviewer 1 Report

Comments and Suggestions for Authors

The authors adequately responded to my remarkes and doubtfulnesses .The new version of  manusript is improved .